# Genetic Architecture Underlying the Metabolites of Chlorogenic Acid Biosynthesis in *Populus tomentosa*

**DOI:** 10.3390/ijms22052386

**Published:** 2021-02-27

**Authors:** Liangchen Yao, Peng Li, Qingzhang Du, Mingyang Quan, Lianzheng Li, Liang Xiao, Fangyuan Song, Wenjie Lu, Yuanyuan Fang, Deqiang Zhang

**Affiliations:** 1National Engineering Laboratory for Tree Breeding, College of Biological Sciences and Technology, Beijing Forestry University, No. 35, Qinghua East Road, Beijing 100083, China; liangchenyao@bjfu.edu.cn (L.Y.); lipeng@bjfu.edu.cn (P.L.); Qingzhangdu@bjfu.edu.cn (Q.D.); Mingyangquan@bjfu.edu.cn (M.Q.); lzlee@bjfu.edu.cn (L.L.); xiaoliang0622@126.com (L.X.); fangyuansong@bjfu.edu.cn (F.S.); wenjielu@bjfu.edu.cn (W.L.); yuanyuanfang@bjfu.edu.cn (Y.F.); 2Key Laboratory of Genetics and Breeding in Forest Trees and Ornamental Plants, Ministry of Education, College of Biological Sciences and Technology, Beijing Forestry University, No. 35, Qinghua East Road, Beijing 100083, China; 3Beijing Advanced Innovation Center for Tree Breeding by Molecular Design, College of Biological Sciences and Technology, Beijing Forestry University, No. 35, Qinghua East Road, Beijing 100083, China

**Keywords:** mGWAS, eQTN, chlorogenic acid, biosynthesis pathway, *Populus*, selective signatures

## Abstract

Chlorogenic acid (CGA) plays a crucial role in defense response, immune regulation, and the response to abiotic stress in plants. However, the genetic regulatory network of CGA biosynthesis pathways in perennial plants remains unclear. Here, we investigated the genetic architecture for CGA biosynthesis using a metabolite-based genome-wide association study (mGWAS) and expression quantitative trait nucleotide (eQTN) mapping in a population of 300 accessions of *Populus tomentosa*. In total, we investigated 204 SNPs which were significantly associated with 11 metabolic traits, corresponding to 206 genes, and were mainly involved in metabolism and cell growth processes of *P. tomentosa*. We identified 874 eQTNs representing 1066 genes, in which the expression and interaction of causal genes affected phenotypic variation. Of these, 102 genes showed significant signatures of selection in three geographical populations, which provided insights into the adaptation of CGA biosynthesis to the local environment. Finally, we constructed a genetic network of six causal genes that coordinately regulate CGA biosynthesis, revealing the multiple regulatory patterns affecting CGA accumulation in *P. tomentosa*. Our study provides a multiomics strategy for understanding the genetic basis underlying the natural variation in the CGA biosynthetic metabolites of *Populus*, which will enhance the genetic development of abiotic-resistance varieties in forest trees.

## 1. Introduction

Chlorogenic acid (CGA) belongs to a class of secondary metabolic products involved in transcriptional activation of the phenylpropanoid pathway under pathogen infection and abiotic stress events [1]. CGA mainly consists of 1-O-CGA, 3-O-CGA, and 5-O-CGA isomers, which have similar biological functions [2]. CGA is widely used in the food and pharmaceutical industries, where it is an effective antibacterial and anti-inflammatory drug in the treatment of cardiovascular diseases and cancer. It is also used as a green protective food additive and can be used to create synthetic antibiotics to reduce drug costs [3]. Importantly, as the main component of phenols, CGA can participate in plant defense reactions in response to various abiotic stresses due to antioxidant and stress resistance of CGA [4]. For example, overexpression of *PbSPMS* (Spermine synthase) contributes to CGA biosynthesis in arabidopsis and improves plant resistance to drought and salt stress [5]. Recent studies have shown that human domestication of lettuce resulted in reduced quinic acid (QA; the precursor metabolite of CGA) and CGA levels in cultivated lettuce, which improves the edible properties of lettuce, but reduces its adaptability to the environment [6]. In addition, CGA is also involved in the adaptability and stability of woody plants under drought and cold conditions [7,8]. Therefore, understanding CGA biosynthesis is significant for innovation in the pharmaceutical industry and improving plant breeding, especially the responses to abiotic stress. However, in perennial woody plants, the regulatory mechanisms of CGA biosynthesis, including how the genetic regulatory network of CGA biosynthesis responds to environmental pressures, remain unclear.

In total, three CGA biosynthetic pathways have been identified in plants that involve identical types of enzyme reactions, including esterification and hydroxylation [9,10]. As the shared precursor of these three pathways, QA has an essential regulatory role in the final abundance of CGA [11]. Meanwhile, QA combined with feruloyl-CoA, was catalyzed by caffeoyl-CoA to synthesize feruloylquinic acid (FQA), the downstream metabolite of CGA biosynthesis. CGA can also synthesize FQA under the catalysis of O-methyltransferase. This suggests the possibility of competition between CGA and FQA in the biosynthesis process [12,13]. Recent studies have found that several catalytic enzymes control CGA biosynthesis. For example, *HQT* (hydroxycinnamoyl CoA quinate hydroxycinnamoyl transferase), a crucial enzyme involved in downstream regulation of the CGA biosynthetic pathway, has been confirmed as the rate-limiting step in many plant species [14]. In the mulberry tree, *HCT4* (hydroxycinnamoyl CoA shikimate/quinate hydroxycinnamoyl transferase 4) increases the CGA content by promoting the biosynthesis of *p*-coumaroylshikimic acid and *p*-coumaroylquinic acid after a frost [10]. Besides, some regulatory factors are reported to be involved in the biosynthesis of CGA by regulating related catalytic enzymes. For example, *bZIP8* (basic-leucine zipper 8) specifically bound the G-box element of *PAL2* (phenylalanine ammonia-lyase 2) 5’-UTR, and overexpression of *bZIP8* inhibited CGA accumulation in *Lonicera japonica* [15]. Five WRKY family members (*WRKYs 38*, *45*, *60*, *89*, and *93*) acted as transcriptional activators for *HCT2* and promoted the biosynthesis of CGA to respond to pathogen infection in poplar [16]. Although the CGA biosynthetic pathway has been partially elucidated in plants, the genetic component of CGA biosynthesis in perennial woody plants remains to be investigated, especially in terms of the complex interaction mechanisms between QA, CGA and FQA-related metabolites involved in growth and abiotic stresses in forest trees.

The application of genome-wide association studies (GWAS) is a powerful tool to identify DNA variant-phenotype associations and is an effective approach to analyze the genetic structure of complex quantitative traits [17]. Metabolome-based GWAS (mGWAS) methods have been widely used in various cash crops and forest trees to decipher the genetic basis of important metabolites, such as salicylic acid, flavonoids and aromatic phenolamides [18,19]. Growing evidence suggests that the effects of causative genes at the transcriptional level provide alternative regulatory mechanisms, ultimately affecting phenotypic variations [20,21]. Expression quantitative trait nucleotide (eQTN) mapping is a powerful approach to establish the link between DNA polymorphisms, gene expression profiles and phenotypic variations, and to provide clues for the mechanisms underlying allelic variations that cause phenotypic variations at the transcriptional level [22,23]. The biosynthetic network of steroidal glycoside alkaloids in tomato was identified through comprehensive utilization of eQTN and mGWAS, which accelerated improvement in fruit quality [24]. Therefore, combining the multiomic strategy of mGWAS and eQTN mapping is conducive to revealing the genetic structure of metabolic traits and construct a comprehensive genetic regulatory network for the biosynthesis of important metabolites in trees.

Here, we quantitatively analyzed the upstream and downstream metabolites in the CGA biosynthetic pathway in 300 accessions of *Populus tomentosa* by high-throughput LC-MS/MS. We deciphered the genetic basis of secondary metabolites related to CGA biosynthesis using mGWAS and eQTN mapping and further clarified the genetic regulatory network of CGA biosynthesis. Furthermore, we characterized the local selection adaptability signatures of CGA biosynthesis-related metabolites across three geographic subpopulations. Our study provides new insights into the genetic regulatory mechanism of CGA biosynthesis in forest trees and accelerates the improvement of genetic resistance in *Populus* varieties.

## 2. Results

### 2.1. Phenotypic Variations of CGA Biosynthesis-Related Metabolic Traits in the Association Population of P. tomentosa

We measured 11 metabolites involved in CGA biosynthesis from 300 accessions of *P*. *tomentosa* in the association population, which were divided into three categories that included QAs, CGAs and FQAs. The coefficients of phenotypic variation (CV) ranged from 45.1% (CGA) to 99.4% (3-O-FQA) (Appendix A), indicating abundant genetic diversity of CGA biosynthesis-related metabolites in the association population of *P*. *tomentosa*. For these 11 metabolite traits, the broad-sense heritability (H^2^) ranged from 0.14 to 0.96, of which CGA, 3-O-*p*-coumaroylquinic acid O-hexoside (3-O-*P*-CGAO-H) and O-FQA displayed an H^2^ value > 0.7 (Table 1), suggesting that the genetic components predominantly contribute to the phenotypic variations. Pearson correlation analysis showed that CGA content was significantly positively correlated with QAs, but negatively correlated with FQAs. In addition, positive phenotypic correlations were detected between CGA and CGA isomers, but not in chlorogenic acid methyl ester (CGAME) (Figure 1A). For example, QA was positively correlated with CGA (*r* = 0.79, *p* = 1.13E-23). In comparison, both 1-O-FQA and O-FQA exhibited negative correlations (*r* = −0.75, *p* = 4.21E-21; *r* = −0.47, *p* = 4.21E-07) with CGA, respectively. These results show the importance of QA and FQA in the CGA biosynthesis pathway.

Significant phenotypic differentiation was detected for the 11 CGA biosynthesis-related metabolic traits among the three subpopulations (Figure 1B and Appendix A). For example, the 5-O-CGA and CGA contents in the north eastern (NE) subpopulation were lower than those in north western (NW) and southern (S) subpopulations. Simultaneously, we found that the content of QA was the highest in the NW subpopulation, followed by the NE and S subpopulations. By contrast, the contents of 1-O-FQA and O-FQA were the highest in the NE subpopulation and gradually decreased in the NW and S subpopulations. The results suggest that CGA biosynthesis-related metabolites have been adapted to different geographical environments by local adaptation under the long process of evolution.

### 2.2. mGWAS for Metabolites Involved in CGA Biosynthesis

To decipher the genetic basis of CGA metabolic traits in *P*. *tomentosa*, mGWAS was performed between 2,415,528 independent SNPs (MAF > 0.05, *r*^2^ < 0.7, heterozygous < 0.95, and missing ratio <20%) and 11 metabolic traits. In total, 219 significant associations, representing 204 unique SNPs, were detected for the 11 metabolic traits (*p* ≤ 4.14E-07), and each SNP explained 3.01–35.4% of the phenotypic variation (R^2^), with an average R^2^ of 13.22% (Appendix A). Next, we annotated 206 genes in the genomic regions of a 10 kb window centered on 125 significant SNPs, and the remaining SNPs were not annotated with any nearby genes (Appendix A). We observed that only 30 of 125 SNPs were located in the coding region of the genes, of which seven were located in the exon regions. In addition, three and 41 SNPs were found in the 3’-UTR and promoter regions of the genes, respectively (Appendix A), suggesting that allelic variations in the noncoding regions may dominate the phenotypic variations. Gene annotation showed that most genes were involved in plant metabolism and cell growth processes, including some genes that have been reported to regulate CGA biosynthesis (Appendix A). For example, SNP 3_10716590 within the promoter region of *Ptom.003G.01386* (encoding the transcription factor *MYB114*) was significantly associated with 5-O-CGA, which regulates CGA biosynthesis via activation of the phenylpropanoid pathway [25]. Another peak association signal for CGA was SNP 18_9299320, which was located in the 2.31 kb downstream region of *Ptom.018G.00976*, and both SNP 18_9299320 and *Ptom.018G.00976* were included in a 5 kb strong LD block (Appendix A). *Ptom.018G.00976* encoded a protein of hydroxycinnamoyl-CoA shikimate/quinate hydroxycinnamoyl transferase (*HCT*), a pivotal rate-limiting enzyme in the CGA biosynthesis pathway.

Among 204 significantly-associated SNPs, 33 and 13 SNPs showed significant additive and dominant effects, respectively, and 12 exhibited both additive and dominant effects (Appendix A). We identified a further 15 SNPs simultaneously associated with two metabolite traits, indicating that these SNPs have pleiotropic effects in the CGA metabolic pathway. For instance, SNP 3_16580246 was located in the 1.40 kb downstream region of *Ptom.003G.01787* (and encoding late embryogenesis abundant hydroxyproline-rich glycoprotein family, *AHL15*), which can affect the accumulation of O-FQA (R^2^ = 12.23%) and CGAME (R^2^ = 10.38%) with different additive and dominant effects, respectively. In addition, a further three SNPs exhibited different genetic effects for different metabolic traits. For instance, SNP 3_18929877 showed a significant additive effect in QA, but a dominant effect occurred in CGA. These results show that the biosynthesis of CGA-related metabolites is a complex genetic process controlled by multiple genes.

### 2.3. Discovery of the Allelic Regulation of Metabolites Involved in CGA Biosynthesis at the Transcriptional Level by eQTN

Because most of the significantly-associated SNPs were located in noncoding sequences, transcriptional regulation was indicated to occur between the metabolome and the genome. We used eQTN to systematically unravel the genetic effects of allelic variation on CGA biosynthesis-related metabolites at the transcriptional level. eQTN mapping was implemented for the expression levels of 147 CGA biosynthetic-related genes (expressed in >80% of the 300 accessions). We detected 874 significant associations (*p* < 4.14E-07), representing 851 unique eQTNs associated with the expression abundance of 109 genes (Figure 2A and Appendix A). Then, we annotated 1066 unique genes (the 10 kb window centered on each significant SNP) from 671 eQTNs, and the remaining eQTNs were located in the intergenic regions. We identified 29 *cis*-eQTNs and 854 *trans*-eQTNs based on the distribution of eQTNs with associated genes, suggesting that *trans*-eQTN is the main model for whole genome-eQTN action in regulating CGA biosynthetic-related genes.

Among these eQTN signals, there were approximately four-fold more eQTNs located in noncoding regions than in coding regions (530 vs. 146), suggesting that variants in noncoding sequences are major regulators of gene expression (Appendix A). Notably, we observed that 17 eQTNs significantly associated with the expression levels of multiple genes (Figure 2A). For example, SNP 5_13235343, located in the 4.13 kb upstream region of *Ptom.005G.01420* (encoding Wiskott-Aldrich syndrome protein 20, *WASP20*) acted as a *trans*-eQTN to affect the transcription level of six genes, which corresponded to four metabolites; 1-O-FQA, CGA, CGAME, and QA (Figure 2C,D). These results suggested that SNP 5_13235343 was an eQTN hotspot with a strong regulatory effect on the expression of several linked genes and affected the expression levels of different genes, and *WASP20* was the master regulator that might indirectly mediate the accumulation of CGA-related metabolites.

### 2.4. Selective Sweep Revealed Genomic Signatures of Adaptive Selection Underlying Metabolites Involved in CGA Biosynthesis

The combination of mGWAS and selective sweep analysis enabled us to understand how CGA biosynthesis and metabolic regulation responds to local environmental pressures. An in-depth trawl of mGWAS, and selective sweep analysis, revealed 16 GWAS signals that overlapped within the potential selective region, and these 16 GWAS signals corresponded to 17 genes. The allelic frequencies of these SNPs were significantly different (≥20% allele frequency difference) among the three geographic regions (Appendix A). For example, SNP 11_3523861 (T/C), located in the 2.69 kb upstream region of *Ptom.011G.00299* (encoding the nucleoside transporter *FUN26*), was associated with 1-O-CGA content and was located in the sweep region on Chr11 (Figure 3A–D). We discovered that all the dominant allelic mutations (C) of SNP 11_3523861 (T/C) were present in the NW subpopulation (23.81%) but completely absent in the NE and S subpopulations (Figure 3E). Similarly, SNP 7_12829919 (T/C) within the promoter region of *Ptom.007G.01300* (encoding the mitochondrial transcription termination factor family protein, *mTERF7*) was located in the sweep region on Chr7, which was associated with QA content (Figure 3A–D). Furthermore, the C-allele in SNP 7_12829919 showed a higher QA value that was strongly selected for by the local environment, resulting in the almost complete elimination of the T-allele in the NW subpopulation (Figure 3F). Therefore, we speculate that the biosynthesis of CGA has been under strong selection to adapt to specific environments in different climatic regions by affecting allelic frequency.

Moreover, we identified 78 eQTN signals located in 80 sweep regions, corresponding to 85 genes in *P*. *tomentosa*, most of which were annotated to plant leaf metabolism and adaptability (Appendix A). For example, a *trans*-eQTN 14_6458025 located in the promoter region of *Ptom.014G.00791* (encoding the BZIP TRANSCRIPTION FACTOR-LIKE PROTEIN 53, *BZIP53*), which could regulate the expression of *Ptom.003G.01787* (encoding the hypothetical protein POPTR_T165600, *NHL15*) and was associated with O-FQA and CGAME. SNP 5_16487452, was located in the promoter region of *Ptom.005G.01695* (encoding REPLICATION FACTOR A 1*, RFA1*), which was a *trans*-eQTN regulating the expression of *Ptom.014G.00243* (encoding interleukin-1 receptor-associated kinase 1, *IRAK1*) that was significantly associated with the O-FQA content. We also saw that the expression of *IRAK1* showed significant differentiation across the three geographic regions, suggesting that gene expression has adapted to local climatic conditions, and this has affected the accumulation of CGA-related metabolites. These genes provide insights into the local environmental adaptability caused by the metabolism of CGA-related metabolites and based on their respective expression levels.

### 2.5. Functional Interpretation of Causal Genes in CGA Biosynthesis

By combining the mGWAS and eQTN results we could update the functional annotation of causal genes involved in the CGA biosynthesis pathway. GO enrichment analysis of candidate genes in the mGWAS and eQTN results showed that approximately one-third of the genes had catalytic activity, hydrolase activity and antioxidant activity (Appendix A), suggesting the potential function of the candidate genes that affected the CGA content. Next, we explored the allelic variants of candidate genes for metabolites related to CGA biosynthesis, including putatively-reported genes and the newly-discovered genes in our work, and constructed a regulatory network of CGA accumulation in *Populus*.

We identified the top-ranking significant SNP 3_18929877 (A/C) within the intron region of *Ptom.003G.01892* (encoding the 60S ribosomal protein L3, *RPL3B*) for QA (*p* = 3.78E-08) and CGA (*p* = 9.27E-08), with significant regional selective signals (Figure 4A,B). In the eQTN mapping, we identified nine genes that regulated the expression level of *RPL3B*, of which the SNP 10_19365920 was a *trans*-eQTN located in the 2.22 kb upstream region of *Ptom.010G.02355* (encoding the MATE EFFLUX FAMILY PROTEIN 1, *MATE1*). SNP 3_9719099 in the 4.28 kb downstream region of *RPL3B* was associated with the expression of *Ptom.006G.00814* (encoding the PATHOGENESIS-RELATED THAUMATIN-LIKE PROTEIN 1, *PR1*) as a *trans*-eQTN. *PR1* was seen to be significantly associated with 1-O-FQA (Appendix A), indicating that *RPL3B* could potentially affect the 1-O-FQA content by controlling the expression of *PR1*. Based on candidate gene analysis, we identified InDel_18929839 (T/TAA) that had high LD with SNP 3_18929877 (*r^2^* = 0.771), which formed a haplotype (A-T/C-TAA) and was significantly associated with QA and CGA (Figure 4D and Appendix A). The allelic variation patterns of SNP 3_18929877 were consistent with two metabolites according to geographic regions, with the dominant allele (C) being fixed in the NW (16.7%) subpopulation but completely absent in the NE and S subpopulations (Figure 4E,F). Further analysis showed that the expression level of *RPL3B* was highly negatively correlated with the abundance of CGA (*r* = −0.618, *p* = 1.73E-11) and QA (*r* = −0.639, *p* = 1.28E-12), and was significantly correlated with the expression level of *PR1* and *MATE1*, respectively (Figure 4G and Appendix A), which partly supported their high phenotypic correlation and regulatory networks.

Another top-ranking significant SNP 9_6689099 (T/C), located in the promoter region of *Ptom.009G.00855* (encoding ethylene-responsive transcription factor *ERF109*), was associated with 1-O-FQA (*p* = 9.05E-09) and CGA (*p* = 1.72E-08), which showed strong selection signals in the NE compared to the NW populations (Figure 4B,C). Concurrently, the expression of *ERF109* was regulated by a lead *trans*-eQTN SNP 1_2206497 within the promoter region of *Ptom.001G.00290* (encoding sigma factor binding protein 1, *SIB1*). In the promoter region of *ERF109*, a lead *cis*-eQTN SNP 4_3592960 regulated *Ptom.001G.00737* expression (encoding an unknown functional protein, *DUF5086*) that was associated with the O-FQA content (Appendix A), indicating that *ERF109* may simultaneously regulate the metabolic levels of CGA, 1-O-FQA, and O-FQA. Association analysis of *ERF109* detected a major haplotype (T-GA-G/C-G-T) that affected the 1-O-FQA and CGA contents (Figure 4D and Appendix A). The SNP 9_6689099 formed a strong LD block (*r^2^* = 0.741, 0.789) with InDel 9_6689118 (GA/G) and SNP 6,689,154 (G/T), which was associated with CGA and 1-O-FQA. However, we found the opposite allelic variation pattern between CGA and 1-O-FQA. For example, the allele frequency of C in SNP 9_6689099 increased from 29.30% in the S to 53.49% in the NW. However, it was absent in the NE subpopulation. The T-allele was almost fixed in the NE subpopulation, where the C and T-alleles corresponded to higher CGA and 1-O-FQA contents in the association population, respectively (Figure 4E,F). Further exploration showed that the *ERF109* expression level was highly correlated with the abundance of CGA (*r* = 0.601, *p* = 1.59E-11) and 1-O-FQA (*r* = −0.569, *p* = 9.51E-10) (Figure 4G and Appendix A), indicating that *ERF109* may be a positive and negative regulator of CGA and 1-O-FQA biosynthesis, respectively. Similarly, we also found a strong correlation in the expression patterns of *ERF109*, *DUF5086*, and *SIB1*, and their mutual interactions provide evidence for the biosynthetic regulatory network of CGA.

Based on the above results, we proposed a main allelic regulatory network of CGA accumulation in *Populus* (Figure 5). We showed that allele variations associated with a high CGA content were prevalent in the drought/rainless NW region (Figure 3E and Figure 4F). Therefore, to explore how environmental selection pressure affects CGA biosynthesis, we further analyzed their expression patterns in the drought-treatment population of *P*. *tomentosa*. Compared with the normal growth stage, the expression levels of *RPL3B* and *MATE1* were significantly downregulated and upregulated, respectively, and the expression of *PR1* was slightly decreased. In addition, the expression levels of *ERF109*, *DUF5086*, and *SIB1* were upregulated to varying degrees (Appendix A). Using a combination of dynamic transcription patterns and correlation analyses of causal genes (Appendix A), we were able to interpret these patterns for the six causal genes and identify a potential new regulatory mechanism for CGA accumulation in *P*. *tomentosa* under stress (Appendix A).

## 3. Discussion

CGA metabolites play indispensable roles in plant defense, redox reactions and abiotic stress responses [1,8]. Here, we analyzed the genetic architecture of metabolites related to CGA biosynthesis in a natural population of *P. tomentosa* using a multiomics method. We combined the variome, transcriptome, and metabolome in CGA biosynthesis-related metabolites to construct genetic regulatory networks that affect CGA biosynthesis and identified potential positive and negative regulatory genes. Our work has laid a theoretical foundation for studying the synthesis and metabolism of plant CGA and its genetic regulation mechanisms in forest trees.

### 3.1. Multiomics Data Analysis of the Metabolites Involved in CGA Biosynthesis in P. tomentosa

In this study, we used the variome, transcriptome, and metabolome of CGA biosynthesis-related metabolites to conduct a multidimensional analysis, which was conducive to the discovery of candidate genes related to the biosynthesis of CGA. The three biosynthetic pathways of CGA have been extensively studied in some herb plants [9,26]. Here, we identified 11 CGA biosynthesis-related metabolites using LC-MS/MS-based metabolic profiling methods across an association population of 300 accessions of *P*. *tomentosa*. The correlation analysis reflected the biological interactions of these metabolites in the CGA biosynthesis process and, to a certain extent, supported the CGA biosynthetic pathway (Figure 1A). In recent years, several genes that regulate CGA biosynthesis have been identified. In *P*. *trichocarpa*, *PtHCT2* significantly affected the metabolic level of CGA, which was regulated through *cis*-eQTLs containing the W-box for *WRKY* binding [16]. Our mGWAS results showed that SNP 18_9299320 (A/C), in the 2.31 kb upstream of *HCT*, was associated with CGA content (Appendix A), highlighting the power of the multi-omics analysis implemented in our study. Interestingly, we also identified a lead *trans*-eQTN SNP 6_14355219 in the 4.59 kb downstream region of *Ptom.006G.01540* (encoding ribulose-bisphosphate carboxylase, *RBCS3B*) that regulated expression levels of *HCT* (Appendix A). In previous studies, *RBCS3B* was shown to contribute to Rubisco accumulation, accelerate the rate of carbon assimilation, and enhance photosynthesis in arabidopsis leaves [27,28]. These studies showed that the expression of *HCT* consisted of a multigene regulatory process in perennial forest trees. In addition, the biosynthetic pathway of CGA also involves *MYB*, *ERF*, *WRKY*, and other transcription factor gene families [14]. Some of their members, including *MYB83*, *MYB114*, *ERF35*, *ERF109* and *WRKY19*, were also identified in our study (Appendix A). For example, *WRKY19* was significantly associated with quinic acid O-di-glucuronic acid (QAO-DCA) content, and its homologous gene, *PtrWRKY19* in *P*. *trichocarpa* inhibited transcription of the *PtoC4H2* promoter containing the conserved W-box element, thus repressing the biosynthesis of CGA [11]. The mGWAS results showed that CGA biosynthesis-related metabolites have a large, multigene genetic basis in *P*. *tomentosa*.

The genetic variation components of the quantitative traits conferred by multiple genes in shared biological pathways were divided into additive, dominant, and epistasis effects [29]. To construct a genetic regulatory network for the complex metabolic traits of forest trees, we conducted in-depth analyses of the genetic effects of significant associated loci. For example, SNP 3_18929877, a pleiotropic locus for QA and CGA, showed significant additive effects (1.29) for QA, but a significant dominant effect (−1.89) for CGA (Appendix A). These findings indicated that these loci affected the metabolic level of CGA biosynthesis-related metabolites through multiple genetic effects, thereby providing a basis for studying the metabolome of *P*. *tomentosa*. In summary, we systematically analyzed the genetic architecture of CGA biosynthesis-related metabolites using mGWAS.

An increasing number of studies have explored the causal loci for phenotypic variation at the transcriptional level. Investigation of eQTNs provides an insight into the gene expression effects of candidate loci and unravels the relationship between genotype and phenotype [30]. Studies based on eQTNs helped us recognize the regulatory factors involved in the CGA biosynthetic pathway at the transcriptional level. We found that the eQTNs located in the noncoding regions were approximately four-fold higher than those in the coding region (78.40 vs. 21.60%) (Appendix A), emphasizing the importance of noncoding sequences in regulating the expression of CGA biosynthetic genes. In addition, we observed that *trans*-eQTNs are more common in the genome, indicating that gene expression is largely regulated by *trans*-eQTNs in our study [31]. More importantly, we identified a total of 34 mGWAS and eQTN overlapping genes (Figure 2B). For instance, *RPL3B*, a major gene affecting QA and CGA metabolism, can also affect the 1-O-FQA content by regulating the expression of *PR1*. This result indicated that these causal genes could not only directly contribute to phenotypic variation at the genomic level, but also affect CGA biosynthesis-related metabolites by regulating the expression of ontology and other genes through the *cis*/*trans*-eQTN. Meanwhile, we identified 17 eQTNs associated with multiple expression genes (Figure 2A), highlighting the possible regulatory role of these eQTN hotspots in CGA biosynthesis. Therefore, eQTN mapping showed another level of interactions among the 11 metabolic traits of *P*. *tomentosa*, linking genetic variation and phenotypic diversification, and enabled the mining of several primary loci in the noncoding transcript. GO enrichment analysis showed that approximately 33.52% of the genes involved protease and hydrolase activities (Appendix A), and this provided genetic information to help clarify the biological function of causative variations underlying some metabolic traits. Our study using multiomics analyses will greatly facilitate large-scale clarification of interactive gene-metabolite annotation and metabolic pathways in plant species.

### 3.2. Local Adaptive Signaling of Metabolites Involved in CGA Biosynthesis among Natural Distributions of P. tomentosa

Previous studies have shown that CGA has significant roles in the response to environmental stress such as drought, cold, strong light and lack of nitrogen and phosphorus, in many plant species [1,32]. The natural population of *P*. *tomentosa* spans a wide climatic range and encounters spatial and geographic structures that determine the effects of environmental factors for phenotypic variation of CGA biosynthesis-related metabolic traits. In the present study, we observed that CGA biosynthesis-related metabolites were significantly different in the three geographic regions of *P*. *tomentosa* (Figure 1 and Appendix A). QA and CGA had the highest content in the NW subpopulation and the lowest content in the NE subpopulation. However, the O-FQA and 1-O-FQA contents were higher in the NE subpopulations but were gradually decreased in the NW and S subpopulations. The metabolite content related to *P*. *tomentosa* CGA biosynthesis varied along an environmental gradient and is likely consistent with climate diversity and latitude differences in the three geographic regions. The NW subpopulation (arid and semi-arid) is located in a high-altitude temperate continental climate, which is dry and cold (average annual temperature, 8.84 °C), with strong ultraviolet radiation and low annual rainfall (544 mm) [33]. Previous studies have shown that the CGA content in *Achillea pachycephala* is significantly increased in response to drought stress [34]. Besides, a previous investigation found that CGA accumulation could enhance the adaptation of apple trees to cold climates [35]. These studies strongly supported the notion that trees can cope with extremely harsh environments and enhance their adaptability to the environment by accumulating abundant CGA. Alternatively, plant viability could be indirectly improved by promoting CGA synthesis via increasing the QA content pathway. The NE subpopulation is located in an area of temperate monsoon climate at high altitude, with an average annual temperature of 12.20 °C, abundant rainfall (606 mm), and fertile soil [33]. Here, environmental stress is reduced, and these trees had low levels of CGA and QA (Figure 1B). The above results indicate that metabolites related to CGA biosynthesis exhibit apparent local adaptability, possibly related to geographic distribution.

A selective sweep is a powerful tool that can be used to identify genomic regions for local adaptation signals in trees and provide additional information about the adaptive mechanisms of subpopulations to their local environment [36]. There were significant differences in the metabolite contents in the three geographic regions (Figure 1B and Appendix A), indicating the genes involved in CGA biosynthesis-related metabolites might experience natural selection pressures. Our study identified 17 and 85 sweeping genes through mGWAS and eQTN analyses, respectively (Appendix A), indicating that these genes regulated the accumulation of metabolites related to CGA biosynthesis and played pivotal roles in adaptation to local environmental pressures. Some of the sweeping genes were involved in plant metabolism and resilience and participated in immune regulation and adversity stress responses. For instance, based on eQTN mapping, sweeping gene *BZIP53* could regulate the expression of *NHL15* and affected the accumulation of O-FQA and CGAME contents (Appendix A). Recent studies have shown that *BZIP53* is a crucial transcription factor regulating CGA biosynthesis in *Eucommia ulmoides*, and can protect plants against biological and pathogenic bacteria [37]. Further observation found that *HCT* was located 4.50 kb downstream of the sweep region 9164001_9168000 on Chr18. Since the LD decay distance was 3–6 kb in *Populus* [38], this indicated that *HCT* might have undergone environmental selection. Noticeably, both the *FUN26* allele for higher 1-O-CGA content and the *mTERF7* allele for higher QA content were fixed in the NW geographic region (Figure 3E,F), suggesting a higher CGA accumulation, as QA and 1-O-CGA have a high positive phenotypic correlation with CGA (Figure 1A). By contrast, the dominant allelic mutation of *ERF109* for 1-O-FQA was found in the NE geographic regions (Figure 4E,F), indicating that the CGA content in this region was relatively low, and the significant negative correlation between FQAs and CGA partially verified this allelic variation pattern (Figure 1A). These results suggest that the dominant alleles in the different genomic regions have experienced diverse selection pressures during local adaptation processes, and exhibited various differentiation patterns. Interestingly, we found that the allelic variation pattern was roughly consistent with the phenotypic differentiation pattern of the metabolite traits, further confirming the functional adaptability of CGA biosynthesis. These results have deepened our understanding of the basis of forest tree adaptability and will improve *Populus* breeding programs in different climates and geographic regions.

### 3.3. Functional Interpretation of Causal Genes in Multiple Regulatory Processes of the CGA Biosynthesis Network

We constructed a genetic regulatory network for CGA biosynthesis by integrating the mGWAS, eQTN and selective sweep results (Figure 5). Typically, QA is an essential precursor metabolite in the three CGA biosynthesis pathways and confers strong resistance to various fungi and viruses [9,39]. Network analysis showed that *RPL3B* was the significant signal associated with QA and CGA and might affect CGA biosynthesis by regulating content of QA and 1-O-FQA, *MATE1* and *PR1* cooperatively to promote regulation. Functional annotation of orthologs of these new *Populus* genes in other model species will improve our understanding of their functions. For instance, after a cold shift, the continuous reduction of the *RPL3B* transcription level causes REI-LIKE protein to accumulate the abundant *RPL3B* paralogue *RPL3A*, increasing nontranslating eukaryotic ribosome content in response to cold stress in arabidopsi*s* [40]. In addition, eQTN mapping supports the role of *RPL3B* in plant stress resistance, and *MATE1* regulated the transcription level of *RPL3B*. Previous studies have confirmed that *MATE1* regulated ABA-sensitivity and increased tolerance to drought by reduced stomatal conductance [41]. Therefore, by affecting the transcription abundance of *RPL3B*, *MATE1* may participate in QA and CGA accumulation. Meanwhile, *RPL3B* can also affect the 1-O-FQA content by regulating the transcription abundance of *PR1*, verifying the biosynthetic pathway to a certain extent. Transcript levels of the homologous gene *AT5G02140* of *PR1* were significantly downregulated in response to cold stress in arabidopsis, and the regulatory network established based on differential genes may be the key to the local adaptation of plants to low temperature [42]. The integration of *RPL3B* with *MATE1* and *PR1* coordinately controlled the content of QA, CGA and 1-O-FQA, providing better insights into the CGA biosynthesis regulatory network in *Populus*.

FQA is an ester of QA and FA, which is closely related to CGA biosynthesis. Previous studies have shown that FQAs were involved in the synthesis of lignin, enhanced the thickness of the cell wall and improved the ability of plants to cope with biological and abiotic stress [43,44]. We identified a major gene, *ERF109*, that was significantly associated with 1-O-FQA and CGA. The homologous *RRTF1* (*REDOX-RESPONSIVE TRANSCRIPTION FACTOR1*) gene, which is part of the *ERF109* family, has been studied in arabidopsis. *RRTF1* regulates redox homeostasis and is involved in local and systemic signal transduction cascades, facilitating reactive oxygen species accumulation in response to abiotic and biotic stress [45]. We also showed that the transcription level of *ERF109* was mainly regulated by *SIB1*, a nuclear protein that can regulate transcription in chloroplasts and may coordinate the response of the plastid genome and nuclear genome to pathogens [46]. Moreover, *ERF109* alters the O-FQA content by adjusting the transcription of *DUF5086*, suggesting a potential function of O-FQA in the process of CGA biosynthesis.

We identified several critical positive and negative regulatory factors based on multiomics strategies and constructed a CGA biosynthesis regulatory network in *Populus* (Figure 5). We identified two key genes, *ERF109* and *RPL3B*, using a combination of mGWAS, eQTN and selection sweep, which showed significant genetic effects. We also uncovered several causal genes and unreported novel functions in this regulatory network of CGA biosynthesis. Two major genes, *RPL3B* and *ERF109*, synergistically controlled the content of QA, CGA and 1-O-FQA. Crucially, the cascade regulatory networks of *MATE1*-*RPL3B*-*PR1* and *SIB1*-*ERF109*-*DUF5086* combined to regulate CGA biosynthesis and affect the O-FQA content. Our results showed the detailed allele regulatory mechanisms that affect CGA accumulation, and provided a better understanding of its regulatory network in *Populus*. Several studies have shown that increased CGA content enhances cell wall thickness and plant antioxidant capacity to adapt to a drought environment in *Populus* [7,47]. Finally, based on the variation pattern and correlation analysis of the causal genes, we proposed a possible transcriptional regulatory network that affects CGA biosynthesis in *Populus* during drought stress (Appendix A). The dynamic changes of the expression abundance of *RPL3B*, *MATE1*, *ERF109* and *SIB1* may promote QA, CGA accumulation and 1-O-FQA content, but biosynthesis of 1-O-FQA was inhibited. *PR1* and *DUF5086* may also change the 1-O-FQA and O-FQA content during CGA biosynthesis. Conversely, since QA is the common precursor of CGA and FQA biosynthesis, competition exists between CGAs and FQAs. Remarkably, the formation of FQA isomers does not depend on CGA but rather on the combination of QA and feruloyl-CoA [48,49]. Therefore, when the content of CGAs increased, the content of FQAs decreased, and the excessive FQAs accumulation may also inhibit CGAs biosynthesis. The differentiation patterns of QA, CGAs, and FQAs in the three geographic regions partially support this conclusion (Figure 1B). Further investigation into the potential genetic basis and gene regulatory networks of these metabolites will aid in the breeding of better varieties through biotechnology or metabolic engineering.

In summary, we combined a multicombination strategy including metabolome, transcriptome and variome data to understand the genetic basis of CGA biosynthesis-related metabolic traits and constructed a genetic regulatory network of metabolites related to CGA biosynthesis using mGWAS and eQTN. Simultaneously, the selective sweep method was used to clarify the local adaptive mechanism of CGA synthesis-related metabolites. Finally, we discovered multiple regulatory mechanisms for CGA biosynthesis. Our study provides a new perspective for studying the genetic regulation of plant secondary metabolites and provides a data-driven foundation for studying the genetic basis of perennial complex plant secondary metabolite biosynthesis.

## 4. Materials and Methods

### 4.1. Plant Material

The association population consisted of 300 accessions of *P*. *tomentosa*, which were asexually propagated via root segments in 2009 in Guan Xian County, Shandong Province, China (36°23′ N, 115°47′ E). Three clonal replications were made using the complete block design method. The association population was randomly selected from a collection of 1047 natural *P*. *tomentosa* individuals [29], representing almost the entire natural distribution of *P*. *tomentosa* (30–40° N, 105–125° E). The association population was divided into three groups that included the southern (S, *n* = 84), northwestern (NW, *n* = 108), and northeastern (NE, *n* = 108) climatic regions of China [33]. In 2016, we sampled fresh leaves from all 300 individuals between 9:00–11:00 AM on a sunny day during the fast-growing season. These were immediately placed in liquid nitrogen and stored at −70 °C until they could be vacuum freeze-dried. The leaf materials were used for metabolic profiling and transcriptome analyses.

In 2019, to perform the drought-tolerant test, we conducted asexually propagation of *P*. *tomentosa* association populations. One hundred-and-twenty genotypes of one-year-old plants were sampled from the asexually propagated *P*. *tomentosa* population, and each genotype was sampled in triplicate. First, during the initial growth stage of the drought-tolerant test population, plants were watered once a week using an automatic irrigation system, and the soil moisture content was maintained above 40%. The water supply was then stopped until the soil moisture content dropped below 10% to evaluate the traits during the drought treatment stage (after four months). Finally, we collected fresh leaves before and after drought stress for transcriptome analysis using the method described above.

### 4.2. Sample Treatment for Metabolite Analysis

The leaves were harvested at the fully expanded stage and freeze-dried for metabolite profiling. The lyophilized leaves were ground, and ~100 mg powder was used in an overnight extraction at 4 °C with 1.0 mL 70% aqueous methanol. Following centrifugation at 10,000× *g* for 10 min, the supernatant was collected and filtered for LC-MS analysis. This enabled quantitative detection of the features of the 11 metabolites involved in CGA biosynthesis [18]. These metabolites were divided into three groups and included the QAs (QA; eudesmoyl quinic acid [EQA]; and QAO-DCA), the CGAs (1-O-CGA; 3-O-CGA; 5-O-CGA; and CGAME), and the FQAs (3-O-*P*-CGAO-H; O-FQA; 1-O-FQA; and 3-O-FQA). The specific analysis steps and parameter settings are described in Appendix A.

### 4.3. Genome Resequencing and Variation (SNP and InDel) Identification

The methods for genome resequencing and genotyping have been described by Xiao et al. [50]. Briefly, the total genomic DNA was extracted from fresh mature leaves of 300 unrelated individuals using the DNeasy plant Mini Kit (Qiagen, Shanghai, China), and a re-sequencing library was constructed according to the sample preparation instructions (Illumina, San Diego, CA, USA). The resequencing of 300 individuals from the *P*. *tomentosa* association population was performed using the Illumina GA II platform (Shanghai Bioarray Co. Ltd., Shanghai, China), and the average sequencing coverage of the raw data was >15-fold. For quality control, short reads and low-quality reads (≤50% of nucleotides with a quality score (Q) < 20) were removed. SOAP v2.20 aligner software [51] was used to align the sequences to the *P*. *tomentosa* reference genome (v 3.0) with the default parameters. We used the Genome Analysis Toolkit (GATK) version 4.0 [52] to perform the single nucleotide polymorphism (SNP) calling, with the following default parameters: SNP: SNP: QD < 5.0 || MQ < 40.0 || FS > 60.0 || SOR > 3.0 || MQRankSum < −12.5 || ReadPosRankSum < −8.0. Finally, 30,432,804 SNPs were identified using Variant Call Format (VCF) tools v4.1 [53] across all individuals.

To control the quality of InDels, only the uniquely-mapped paired-end reads were used to perform InDel calling, using GATK version 4.0 and SAMTOOLS version 1.3.1 [52,54] with the default parameters. The original InDel data was filtered using the VCFtools v4.1 [53] with InDel size > 1 bp, quality score (Q) > 20, missing rate ≤ 0.20, and minor allele frequency (MAF) > 0.05.

### 4.4. Population Genetic Structure Analysis

We filtered SNPs with a heterozygosity frequency > 95%, and defined SNPs with a MAF > 0.05 and an *r^2^* of linkage disequilibrium (LD) < 0.7 as the standard SNP. In total, 2,415,528 SNPs were selected according to the above criteria and were used in the population structure analysis using Admixture v1.3.0 software [55]. We used the Bayesian clustering program fastStructure [56] to calculate the variation level of K (K = 1–10) in the *P*. *tomentosa* association population, and the minimum CV error value appeared at K = 3 (the optimum population value for K).

### 4.5. mGWAS

The mixed linear model (MLM) in TASSEL v5.0 [57] was applied to analyze the properties of the 11 metabolites, with population structure (Q) and the coefficient of kinship (K) considered covariates. The K matrix was generated using TASSEL v5.0 [57]. After removal of low-frequency SNPs (MAF < 5%, missing >20%, *r*^2^ > 0.7, and heterozygous alleles >95%), the remaining SNPs were used to perform the GWAS using TASSEL v5.0 [57]. The genome-wide significance thresholds for *p*-value ≤ 4.14E-07 (*p* = 1/n; n represents the effective number of independent SNPs) for GWAS were modified by a Bonferroni correction. Following the GWAS, Manhattan plots were drawn using the qqman package in R [58]. Finally, genes that overlapped with a 10 kb window centered on each significant SNP were defined as candidate genes.

### 4.6. RNA Extraction and RNA Sequencing

We extracted total RNA from the mature leaves using the Qiagen RNeasy kit (Qiagen) according to the manufacturer’s protocols. After purification and DNA digestion with RNase-free DNase (Qiagen), high-quality RNA was determined by Nanodrop Spectrophotometer (Thermo Scientific, Waltham, MA, USA) and a Qubit 2.0 Fluorometer (Invitrogen, Life Technologies, Waltham, MA, USA). This RNA was used to construct the cDNA library according to the sample preparation instructions (Illumina). The method for obtaining the transcriptome data is described in detail in Appendix A.

### 4.7. eQTN Analysis

Transcripts with a miss rate of < 20% in 300 individuals were retained for eQTN analysis. The log10 (1 + FPKM) was used to represent the expression levels of trait-associated genes. Then, eQTN analysis was performed using TASSEL v5.0 with K and Q as covariates [57]. The genome-wide significance threshold was set to a *p*-value ≤ 4.14E-07 (1/n). The detected eQTNs located in the 10 kb window around the expressed gene were defined as *cis*-eQTNs, and the remainder were treated as *trans*-eQTNs. The Circos plot of transcriptional regulatory genes involved in CGA biosynthesis were drawn using TBtools [59]. We used the online tool of Jvenn (http://jvenn.toulouse.inra.fr/ (accessed on 23 October 2020) to draw the Venn diagram of all candidate genes in the GWAS and eQTN results.

### 4.8. Screening for Selective Sweeps

To determine the genomic region of local adaptation among the three geographical subpopulations of *P*. *tomentosa*, we first calculated the genetic diversity level (π) and genetic differentiation (F_ST_) of a 2 kb window between each pair of the three subpopulations (NW, NE, and S) [24]. If the π value of the window was <0.001 in a population, the window was removed. We comparatively analyzed the ratios of genetic diversity between the NE and NW subpopulations (π_NE_/π_NW_), the NW and S populations (π_NW_/π_S_), and the NE and S population (π_NE_/π_S_). Then, we selected windows with the top 5% of ratios and the maximum absolute value of Fst as candidate regions for further analysis [24]. Finally, the region defined by the π and F_ST_ sweeps was regarded as the sweep region, and windows spaced 5 kb apart were merged into a sweep region. If genes overlapped with a sweep region, they were considered to be selective sweep genes.

### 4.9. Gene Annotation and Gene Ontology (GO) Enrichment Analysis

All genes identified by GWAS, eQTN, and selective sweeps were annotated according to the *P. tomentosa* genome. First, the Kyoto Encyclopedia of Genes and Genomes (KEGG) and the Nonredundant Protein Sequence Database (NR) enrichments were determined using the respective (https://www.kegg.jp/ (accessed on 26 August 2020)) and (https://ftp.ncbi.nlm.nih.gov/blast/db/FASTA/ (accessed on 26 August 2020)) websites. Then, we used POPGENE (http://popgenie.org/ (accessed on 26 August 2020)) and TAIR (https://www.arabidopsis.org/ (accessed on 26 August 2020)) to annotate candidate homologous genes of *P. tomentosa* in *Populus trichocarpa* and arabidopsis. Finally, we used TBtools [59] for GO enrichment analysis of all the genes. Significant GO annotations were determined at a false discovery rate (FDR) threshold of <0.05.

### 4.10. Candidate Gene-Based Association Analysis

We extracted InDels and SNP variations from the candidate genes using Vcftools, with the following criteria: MAF > 0.05, *r*^2^ < 0.7, heterozygous < 0.95, and missing ratio < 20%. Then, to consider the population structure and relatedness of individuals, candidate gene-based association analysis was performed using the MLM in TASSEL v5.0 [57] to detect potential causal variations, including InDels and SNP markers, associated with phenotypic variations of CGA-related metabolites. We used the FDR method to correct multiple tests using the QVALUE package in R. The significance level of the association result was *p* < 0.001, *Q* < 0.05.

## 5. Conclusions

Taken together, the system genetic strategy based on association studies and expression profiling methodologies enabled us to investigate comprehensively the genetic architecture (additive and dominant effects) of CGA biosynthesis-related metabolites in *Populus*. Here, the synthetic analysis of the result of mGWAS, eQTN and selective sweep identified six causative genes and thus constructed an allele regulatory network underlying CGA biosynthesis. Of these, two major genes *RPL3B* and *ERF109* were the central nodes of the regulatory network and showed significant regional selective signals, which coordinated the accumulation of CGA with the novel genes. In addition, the transcriptional regulation mechanism in promoting the CGA accumulation responding to abiotic stress was verified in the drought-treatment population of *P. tomentosa*. Notably, we identified 16 GWAS signals and 78 eQTNs located in selective sweep regions of the genome, which were highly differentiated in allele frequency among the three geographical populations, revealing the local adaptability of CGA biosynthesis in *Populus*. This study reveals a multiomics view of the CGA biosynthesis pathway in *Populus*, as well as providing insights into marker-assisted breeding to select high resistance elite variety of forest tree.

## Figures and Tables

**Figure 1 ijms-22-02386-f001:**
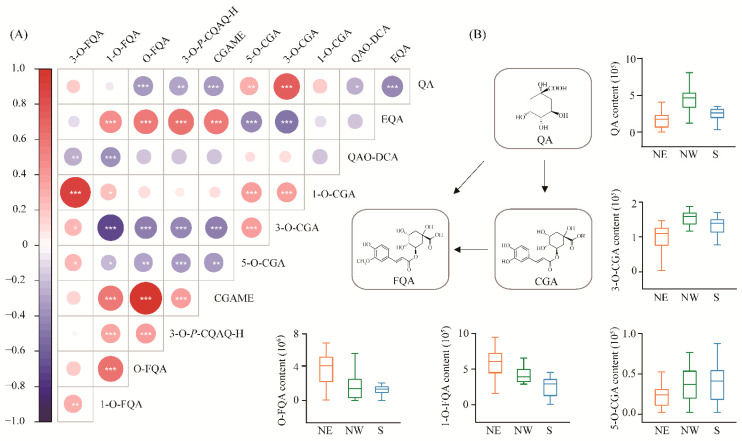
Correlation matrix and subpopulation differentiation of metabolic traits involved in CGA biosynthesis. (**A**) The features of 11 metabolites were quantitatively detected by high-throughput LC-MS/MS analysis, and the correlation between metabolites is shown. The 11 metabolites were divided into three groups, including the QAs (quinic acid [QA]; eudesmoyl quinic acid [EQA]; and quinic acid O-di-glucuronic acid [QAO-DCA]), CGAs (1-O-chlorogenic acid [1-O-CGA]; 3-O-chlorogenic acid [3-O-CGA]; 5-O-chlorogenic acid [5-O-CGA]; and chlorogenic acid methyl ester [CGAME]), and FQAs (3-O-*p*-coumaroylquinic acid O-hexoside [3-O-*P*-CGAO-H]; O-feruloylquinic acid [O-FQA]; 1-O-feruloylquinic acid [1-O-FQA]; and 3-O-feruloylquinic acid [3-O-FQA]). Blue indicates a negative correlation, red indicates a positive correlation, and the size of each circle is proportional to the strength of the correlation (*, *p* < 0.05; **, *p* < 0.01; ***, *p* < 0.001); (**B**) The metabolic conversion pathway of the three types of metabolites is described. The box plot shows the phenotypic differentiation of five metabolites in three geographic regions, and yellow, green, and blue represent the northeastern (NE), northwestern (NW), and southern (S) regions of China, respectively.

**Figure 2 ijms-22-02386-f002:**
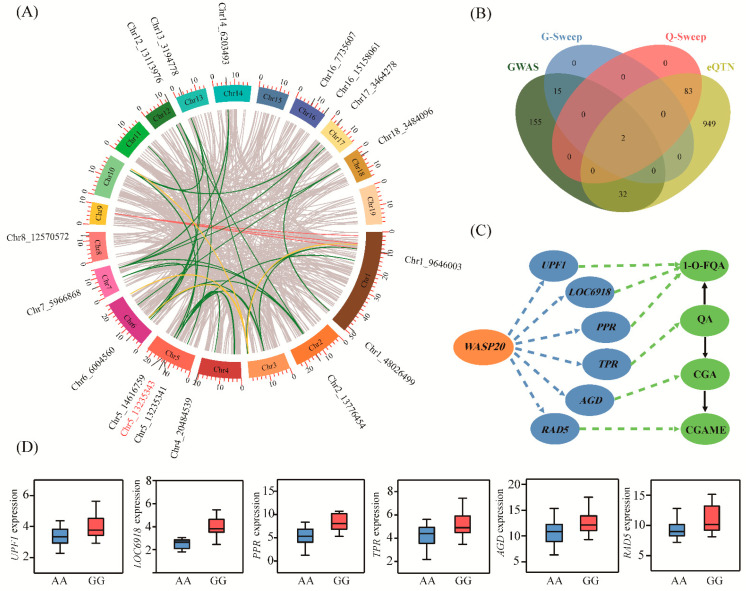
The transcriptional regulatory networks of alleles within candidate genes related to the CGA biosynthesis pathway. (**A**) The Circos plot shows the transcriptional regulation of 1066 genes involved in CGA biosynthesis. The outer circle represents the chromosome, and interior lines represent pairwise interactions. The green line represents the transcriptional regulatory network of the 17 expression quantitative trait nucleotide (eQTN) hot spots corresponding to genes, and their positions on the chromosomes are also marked. The yellow and blue lines represent the expression regulation modes of *Ptom.003G.01892* and *Ptom.009G.00855* in the eQTN mapping, respectively. (**B**) Cross-genes between four independent datasets. The green and yellow modules represent genes identified in genome-wide association study (GWAS) and eQTN analysis, respectively. The blue and pink modules respectively indicate the selective sweep genes in the two analysis results above. (**C**) The genetic regulation mediated by *WASP20* involved in CGA biosynthesis. *Ptom.003G.01299*, *UPF1*; *Ptom.003G.01300*, *LOC6918*; *Ptom.004G.01447*, *PPR*; *Ptom.010G.02293*, *TPR*; *Ptom.001G.01742*, *AGD*; *Ptom.004G.00379*, *RAD5*. The blue and green circles represent expressed genes and metabolites regulated by *WASP20*, respectively. The blue and green dashed lines represent the regulatory networks analyzed in eQTN and GWAS, respectively, and the black lines represent the conversion relationships between metabolites. (**D**) Box diagrams for the eQTN (SNP 5_ 13235343) in the *WASP20* associated with the expression abundance of six genes. The central line indicates the median, and the box limits are the upper and lower quartiles, respectively. The pink and blue boxes represent dominant alleles and inferior alleles, respectively.

**Figure 3 ijms-22-02386-f003:**
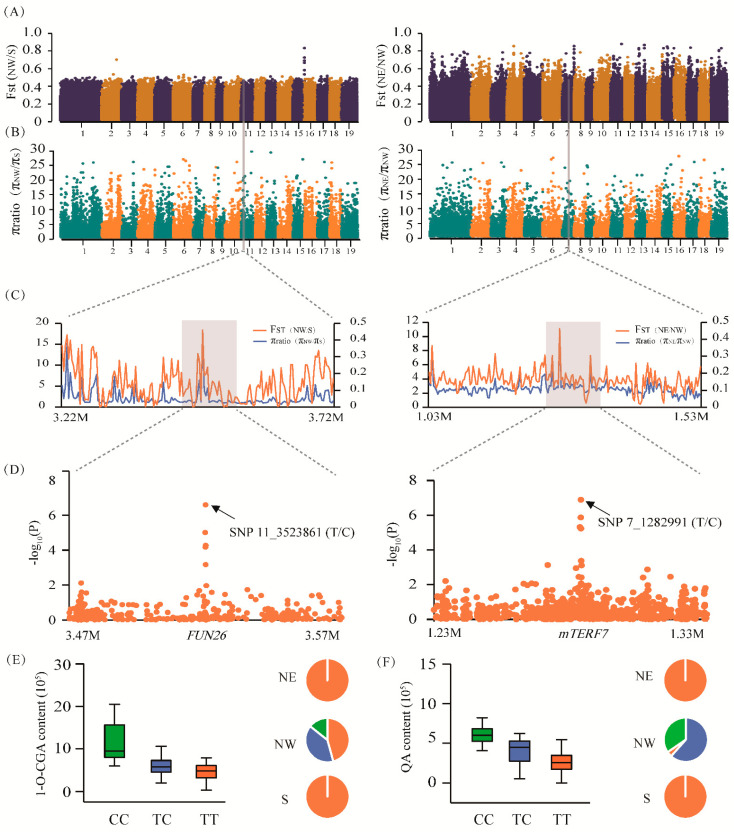
*FUN26* and *mTERF7* show strong adaptation to the local environment in *Populus tomentosa.* (**A**) F_ST_ bar plot for the groups NW vs. S, and NE vs. NW in the whole genome. (**B**) π ratio bar plot for the groups NW vs. S, and NE vs. NW in the whole genome. (**C**) The genomic regions of *Ptom.011G.00299* (*FUN26*; left) and *Ptom.007G.01300* (*mTERF7*; right) both showed strong selective sweep signals in the NW subpopulation. We used a 2 kb sliding window to plot the θπ ratio and Fst value. The grey background modules represent the significant selective region. (**D**) The Manhattan plot shows the regional correlation of 1-O-chlorogenic acid (left) and quinic acid (right) content from a genome-wide association study that coincides with the selected region. (**E**,**F**) The box plots (left) are the genotypic effects of SNP 11_3523861 and SNP 7_12829919 with 1-O-chlorogenic acid and quinic acid content, respectively. The pie chart (right) shows their allelic frequency in the three geographic regions of NE, NW, and S.

**Figure 4 ijms-22-02386-f004:**
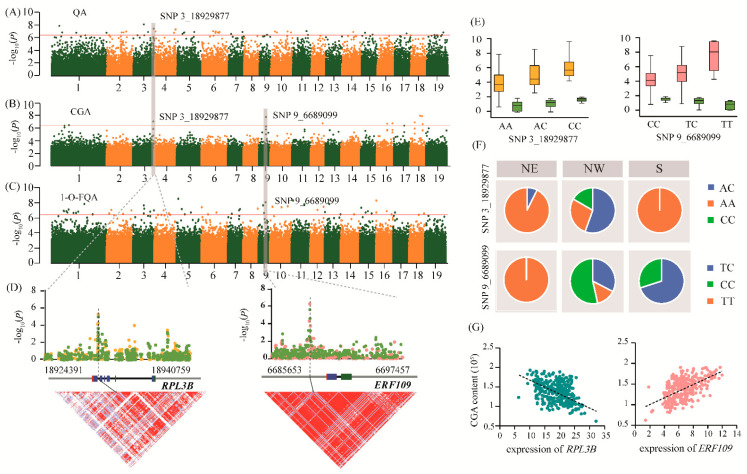
Two crucial genes, RPL3B and ERF109, coordinately regulate CGA biosynthesis. (**A**–**C**) Manhattan plots of the genome-wide association study with quinic acid, chlorogenic acid, and 1-O-ferulylquinic acid content, respectively. The horizontal red line indicates the significance threshold (*p* = 4.14E-07). The gray background modules indicate the signal containing the candidate genes. (**D**) The association analysis between the candidate genes Ptom.003G.01892 (RPL3B; left) and Ptom.009G.00855 (ERF109; right) with target traits. The green, yellow, and pink dots represent chlorogenic acid, quinic acid, and 1-O-ferulylquinic acid, respectively. The gene structures are shown in the middle (red, 5’-UTR; blue, exon; black, intron; green, 3’-UTR; grey, flanking sequence of the gene). Pairwise LD between the lead SNPs (SNP 3_18929877 and SNP 9_6689099) and the surrounding SNPs is shown at the bottom. (**E**) The box plot shows the genotypic effects of SNP 3_18929877 and SNP _6689099 with the target metabolite content, respectively. The color of each of the metabolites is consistent with the above. (**F**) The allele frequencies of two pleiotropic SNPs across the three geographical regions in a natural population of *P. tomentosa*. (**G**) The scatter plot shows the correlation between the expression abundance and chlorogenic acid content of RPL3B (left) and ERF109 (right).

**Figure 5 ijms-22-02386-f005:**
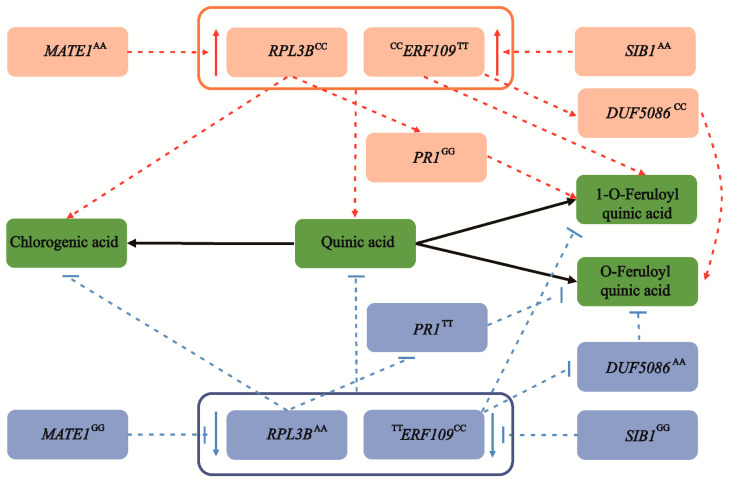
The allelic genetic regulatory network mediated by six causal genes involved in CGA biosynthesis. Two major genes, *RPL3B* and *ERF109*, are significantly associated with the chlorogenic acid content, and coordinate the accumulation of quinic acid, chlorogenic acid, and 1-O-ferulylquinic acid. In the eQTN mapping, *MATE1* and *SIB1* act as the lead *trans*-eQTN to mediate the expression of *RPL3B* and *ERF109*, respectively, and are indirectly involved in the regulation of chlorogenic acid biosynthesis. Meanwhile, *RPL3B* can also alter the content of 1-O-FQA by affecting the expression level of *PR1*. Similarly, *ERF109* also affects the biosynthesis of O-FQA by regulating the abundance of *DUF5086*. The superscript of the gene represents the allele of the causal SNP within the gene. The red and blue circles are the dominant alleles and inferior alleles, respectively. However, as a pleiotropic gene, *ERF109* shows different allelic variation patterns in chlorogenic acid and 1-O-ferulylquinic acid. The dotted lines represent the regulatory network by eQTN and GWAS analyses, and the black solid line represents the transformation relationship between metabolites. The red and blue arrows represent an increase and decrease, respectively.

**Table 1 ijms-22-02386-t001:** Summary of significant associations of SNPs within 11 metabolites.

Metabolite Traits ^a^	Heritability	No. SNPs ^b^	No. Genes ^c^	Additive Effects	Dominant Effects	*R* ^2^
QA	0.142991	37	44	0.92~2.16	−0.71~0.07	0.04~0.30
EQA	0.190704	12	8	1.33~1.37	−0.53~0.13	0.21~0.35
QAO-DCA	0.200318	7	4			0.03~0.08
1-O-CGA	0.359949	13	5	0.86~1.48	−1.53~−0.10	0.11~0.31
3-O-CGA	0.957155	12	13	0.21~1.28	−1.89~−1.16	0.17~0.25
5-O-CGA	0.337585	10	11	1.71	−2.72	0.16~0.26
CGAME	0.818917	23	23	1.41~1.86	−2.1~−1.16	0.04~0.19
3-O-*P*-CGAO-H	0.189497	10	11	0.44~1.60	−0.42~−0.10	0.21~0.25
O-FQA	0.876821	20	17	0.67~1.78	−2.09~0.23	0.03~0.30
1-O-FQA	0.573541	60	61	0.97~1.45	−0.44~0.06	0.05~0.25
3-O-FQA	0.304793	15	21	0.51 0.64	−1.47~0.39	0.06~0.21

^a^ The metabolites are quinic acid, eudesmoyl quinic acid, quinic acid O-di-glucuronic acid, 1-O-chlorogenic acid, 3-O-chlorogenic acid, 5-O-chlorogenic acid, chlorogenic acid methyl ester, 3-O-*p*-coumaroyl quinic acid O-hexoside, O-feruloylquinic acid, 1-O-feruloylquinic acid, 3-O-feruloylquinic acid, in order. ^b^ The number of significant association SNPs. ^c^ The number of possible candidate genes.

## Data Availability

The genome re-sequencing raw data of 300 *P. tomentosa* individuals has been submitted to the Genome Sequence Archive in the BIG Data Center, Beijing Institute of Genomics (BIG), Chinese Academy of Sciences (CAS) under accession number CRA000903, and is publicly accessible at http://bigd.big.ac.cn/gsa/. The sequence information of six candidate genes (*RPL3B*; *ERF109*; *MATE1*; *PR1*; *SIB1*; *DUF5086*) were deposited in GenBank under accession numbers MW540507 to MW540512.

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
