# Peer review of "Genetic Architecture Underlying the Metabolites of Chlorogenic Acid Biosynthesis in Populus tomentosa"

_ijms, 2021, doi:10.3390/ijms22052386_

Round 1

Reviewer 1 Report

The authors use metabolomics and genomics to elucidate the biosynthesis pathway of chlorogenic acid (CGA) in Populus tomentosa. This is a very interesting study and a well-written manuscript. The authors were able to identify SNP and eQTN associated with CGA and up- and downstream metabolites. Furthermore, a few SNPs were found in CGA biosynthesis genes. Based on their metabolite, SNP, and eQTN analysis the authors propose a model of the regulatory network for CGA biosynthesis in Populus tomentosa.

Two minor comments:

I think it would be helpful to give the full name of a gene at the first instance of use. For example, line 45 PbSPMS (Spermine synthase).

Line 611 Perhaps better to say “fully expanded” rather than “fully ripe” when talking about leaves.

Author Response

We sincerely appreciate Reviewer’s efforts and constructive suggestions for our manuscript. We have made our best effort to address the following concerns point by point in the newly revised version. Thanks again for the reviewer’s comprehensive comments and valuable suggestions to improve the manuscript.

Point 1: I think it would be helpful to give the full name of a gene at the first instance of use. For example, line 45 PbSPMS (Spermine synthase).

Response to Point 1: We sincerely thank the reviewers for their detailed and constructive comments. In the newly revised version, we have added the full name of a gene at the first instance of use. As follows: line 45 PbSPMS (Spermine synthase); line 66-67 HQT (hydroxycinnamoyl CoA quinate hydroxycinnamoyl transferase); line 69-70 HCT4 (hydroxycinnamoyl CoA shikimate/quinate hydroxycinnamoyl transferase 4); line 72 bZIP8 (basic-leucine zipper 8); line 73 PAL2 (phenylalanine ammonia-lyase2); line 548-549 RRTF1 (REDOX-RESPONSIVE TRANSCRIPTION FACTOR 1).

Point 2: Line 611 Perhaps better to say “fully expanded” rather than “fully ripe” when talking about leaves.

Response to Point 2: We sincerely thank the reviewers for their detailed and constructive comments. We agree with reviewer1, in the newly revised version, we have changed “fully ripe” to “fully expanded”.

Reviewer 2 Report

I would like to thank the editor and authors for giving me the opportunity to review this manuscript. The manuscript “Genetic Architecture Underlying the Metabolites of Chlorogenic Acid Biosynthesis in Populus Tomentosa” studies the genetic regulatory network of CGA biosynthesis pathways in Populus Tomentosa using metabolite-based genome-wide association study and expression quantitative trait nucleotide mapping. The study provides a new perspective for studying the genetic regulation of plant secondary metabolites biosynthesis. Generally, the topic is highly relevant from both scientific and practical points of view and the problem is well mentioned, the manuscript is well organized with very clear and fully defined illustrations however, some minor edits still required.

  • Poor language and style of writing at some places as:

Line 43; “Importantly, as the main component of phenols, CGA can participate in plant defense reactions in response to various abiotic stresses due to its antioxidant and stress resistance” do you mean induction of stress resistance in plants by CGA or what?? please re-phrase.

Line 46; “Recent studies have shown that human selection resulted in reduced quinic acid (QA; the precursor metabolite of CGA) and CGA levels in cultivated lettuce, which improves the edible properties of lettuce, but reduces its adaptability to the environment” do you mean human selection for lettuce, please rephrase.

Line 602; “In 2019, to perform the drought-tolerant test, genotypes from 120 one-year-old plants …………” do you mean 120 genotypes one-year-old plants?

  • Please mention the definition of “PbSPMS” in line 45.
  • Please indicate which tool or software used in doing the circular and Venn diagrams in Fig 2.
  • Please indicate the time of sampling 9-11 A.M or P.M. in line 598?
  • The results and the discussions were presented in a descriptive way, I encourage the authors to make an integrative discussion of all the results.

For these reasons, I recommend the MS for publication after doing the required revision for the manuscript particularly the English language editing.

Author Response

We appreciate the Reviewer’s detailed suggestions and forward-looking comments on our manuscript. Following your valuable comments, we have incorporated most of the changes in this revised version. Thank you for all the efforts.

Point 1: Poor language and style of writing at some places as:

1.1 Line 43; “Importantly, as the main component of phenols, CGA can participate in plant defense reactions in response to various abiotic stresses due to its antioxidant and stress resistance” do you mean induction of stress resistance in plants by CGA or what?? please re-phrase.

1.2 Line 46; “Recent studies have shown that human selection resulted in reduced quinic acid (QA; the precursor metabolite of CGA) and CGA levels in cultivated lettuce, which improves the edible properties of lettuce, but reduces its adaptability to the environment” do you mean human selection for lettuce, please rephrase.

1.3 Line 602; “In 2019, to perform the drought-tolerant test, genotypes from 120 one-year-old plants …………” do you mean 120 genotypes one-year-old plants?

Response to Point 1.1: We are very grateful for the reviewer’s detailed comments. This sentence may confuse you. Line 43-45, what we want to express in this sentence in the manuscript is that CGA induces stress resistance in plants.

In the newly revised version, we have corrected the language error of this sentence, and the corrected sentence is: Importantly, as the main component of phenols, CGA can participate in plant defense re-actions in response to various abiotic stresses due to antioxidant and stress resistance of CGA.

Response to Point 1.2: We sincerely appreciate the reviewer for the detailed and valuable comment. Line 48-51, what we want to express in this sentence is: In the process of lettuce domestication, artificial selection of lettuce with better flavor resulted in the decrease of QA and CGA content in cultivated lettuce, and reduced the environmental adaptability of cultivated lettuce.

In the newly revised version, we have corrected the language error of this sentence, and the corrected sentence is: Recent studies have shown that human domestication of lettuce resulted in reduced quinic acid (QA; the precursor metabolite of CGA) and CGA levels in cultivated lettuce, which improves the edible properties of lettuce, but reduces its adaptability to the environment.

Response to Point 1.3: We sincerely appreciate the reviewer for the detailed and valuable comment. Line 607-609, Since the sentence is more general, in the newly revised version, we describe the process of establishing a drought-tolerant test population of P. tomentosa in more detail.

The corrected sentence is: In 2019, to perform the drought-tolerant test, we conducted asexually propagate of P. tomentosa association populations, 120 one-year-old genotypes plants were sampled from the asexually propagated population of P. tomentosa, and each genotype was sampled in triplicate.

Point 2: Please mention the definition of “PbSPMS” in line 45.

Response to Point 2: We very appreciate the reviewer’s professional comments. Line 45, we have added the definition of PbSPMS (Spermine synthase) in the newly revised version.

Point 3: Please indicate which tool or software used in doing the circular and Venn diagrams in Fig 2.

Response to Point 3: We very appreciate the reviewer’s professional comments. Line 684-687, we proved the software for drawing Circos plot and the online website for drawing Venn diagrams.

The added sentence is: The Circos plot of transcriptional regulatory genes involved in CGA biosynthesis were drawn using TBtools [59]. We used the online tool of Jvenn (http://jvenn.toulouse.inra.fr/) to draw the Venn diagram of all candidate genes in the GWAS and eQTN results.

Point 4: Please indicate the time of sampling 9-11 A.M or P.M. in line 598?

Response to Point 4: We appreciate the reviewer’s attentive comments, we made a small mistake. Line 604, the sampling time should be 9-11 A.M. In the newly revised version, we have corrected this error.

The corrected sentence is: In 2016, we sampled fresh leaves from all 300 individuals between 9:00–11:00 AM on a sunny day during the fast-growing season.

Point 5: The results and the discussions were presented in a descriptive way, I encourage the authors to make an integrative discussion of all the results.

Response to Point 5: We very appreciate the reviewer for the detailed and valuable comment. We know that there are still some shortcomings in the results and discussion part of our manuscript, and there is a lack of comprehensive discussion. Therefore, we have added the conclusion (Line 721-736) part of the manuscript in the newly revised version. A comprehensive analysis of the technical methods and achievements of this study has enhanced the readability of the article.

Reviewer 3 Report

The paper touches many interesting and important problems related to procedures of CGA biosynthesis pathways. Chlorogenic acid (CGA) is very important in defense response, immune regulation, and the response to abiotic stress in plants. Regarding regulation CGA biosynthesis pathways can in a future powerful tool for understanding the genetic basis underlying the natural variation in the CGA biosynthetic metabolites of Populus as a model species, which will enhance the genetic development of abiotic-resistance varieties in forest trees.

The paper is written logically to describe the process of methodology and is consistent in the structure. The introduction is the basis of the analytical process and the obtained results were discussed in the relevant literature. The conclusions are too general, but the study provides a new perspective for studying the genetic regulation of plant secondary metabolites and provides a data-driven foundation for studying the genetic basis of perennial complex plant secondary metabolite biosynthesis. I would like to suggest adding the section of conclusions to make results more visible to the audience.

I appreciate huge work to perform the analysis and preparation of the manuscript.

Author Response

Response to Reviewer’s comments: We appreciate the Reviewer’s detailed suggestions and forward-looking comments on our manuscript. Following your valuable comments, we have added the conclusions to this revised edition. Thank you for your efforts.

Line 721-736, the conclusion part is as follows: Taken together, the system genetic strategy based on association studies and expression profiling methodologies enabled us to investigate comprehensively the genetic architecture (additive and dominant effects) of CGA biosynthesis related metabolites in Populus. Here, the synthetically analysis of the result of mGWAS, eQTN and selective sweep identified six causative genes and thus constructed an allele regulatory network underlying CGA biosynthesis. Of these, two major genes RPL3B and ERF109 were the central nodes of the regulatory network and showed significant regional selective signals, which coordinated the accumulation of CGA with the novel genes. In addition, the transcriptional regulation mechanism in promoting the CGA accumulation responding to abiotic stress was verified in the drought-treatment population of P. tomentosa. Notably, we identified 16 GWAS signals and 78 eQTNs located in selective sweep region in genome, which highly differentiated in allele frequency among the three geographical populations, revealing the local adaptability of CGA biosynthesis in Populus. This study reveals a multi-omics view of the CGA biosynthesis pathway in Populus, as well as provides insights into marker-assisted breeding to select high resistance elite variety of forest tree.
